



Atmospheric
Measurement
Techniques

# Neural network radiative transfer for imaging spectroscopy

**Brian D. Bue**[1], **David R. Thompson**[1], **Shubhankar Deshpande**[2], **Michael Eastwood**[1], **Robert O. Green**[1], **Vijay Natraj**[1], **Terry Mullen**[3], **and Mario Parente**[3]

[1]Jet Propulsion Laboratory, California Institute of Technology, Pasadena, CA, USA
[2]The Robotics Institute, Carnegie Mellon University, Pittsburgh, PA, USA
[3]University of Massachusetts, Amherst, MA, USA

**Correspondence:** Brian D. Bue (bbue@jpl.nasa.gov)

**Abstract.** TS1 CE1 Visible–shortwave infrared imaging spectroscopy provides valuable remote measurements of Earth's surface and atmospheric properties. These measurements generally rely on inversions of computationally intensive radiative transfer models (RTMs). RTMs' computational expense makes them difficult to use with high-volume imaging spectrometers, and forces approximations such as lookup table interpolation and surface–atmosphere decoupling. These compromises limit the accuracy and flexibility of the remote retrieval; dramatic speed improvements in radiative transfer models could significantly improve the utility and interpretability of remote spectroscopy for Earth science. This study demonstrates that nonparametric function approximation with neural networks can replicate radiative transfer calculations and generate accurate radiance spectra at multiple wavelengths over a diverse range of surface and atmosphere state parameters. We also demonstrate such models can act as surrogate forward models for atmospheric correction procedures. Incorporating physical knowledge into the network structure provides improved interpretability and model efficiency. We evaluate the approach in atmospheric correction of data from the PRISM airborne imaging spectrometer, and demonstrate accurate emulation of radiative transfer calculations, which run several orders of magnitude faster than first-principle models. These results are particularly amenable to iterative spectrum fitting approaches, providing analytical benefits including statistically rigorous treatment of uncertainty and the potential to recover information on spectrally broad signals.

*Copyright statement.* The author's copyright for this publication is transferred to the Jet Propulsion Laboratory, California Institute of Technology.

## 1 Introduction

Remote visible–shortwave infrared (VSWIR) imaging spectroscopy, also known as hyperspectral imaging, is a powerful approach to map the composition, health, and biodiversity of Earth's ecosystems (ESAS, 2018). Remote sensing of the solar-reflected spectrum from 380 to 2500 nm reveals physics and chemistry of many processes in Earth's surface–atmosphere system (Schaepman et al., 2009), including terrestrial plant health and traits (Asner et al., 2017; Ustin et al., 2004); biodiversity (Jetz et al., 2016); the condition and composition of aquatic, benthic, and nearshore ecosystems (Fichot et al., 2015; Hochberg, 2011); geology (Swayze et al., 2014); and trace greenhouse gases (Frankenberg et al., 2016). While Earth scientists aim to measure these geophysical variables, remote sensors can only measure the incident light at the sensor. Inferring geophysical properties requires inverting the measurement with a physical model – typically one that accounts for both absorption and scattering by the atmosphere, and the fraction of light reflected from the surface at each wavelength (Schaepman-Strub et al., 2006).

Radiative transfer models (RTMs) such as DISORT (Stamnes et al., 1988) are a critical component of such models, and form the core of common spectroscopy analysis codes including ACORN (Kruse, 2004), ATCOR (Richter and Schlapfer, 2002), FLAASH (Perkins et al., 2012), ATREM (Gao et al., 1993), and others (Gao et al., 2000, 2007; Thompson et al., 2015). The RTM posits a strati-

fied atmosphere populated by atmospheric gases at appropriate concentrations and temperatures, and solves the general equations of radiative transport based on a known solar input and presumed surface. This is an intensive computation, requiring special care for modern high-volume imaging spectrometers that acquire thousands or millions of spectra per second.

Because imaging spectrometers produce too much data to analyze each measurement with an independent RTM, investigators use RTMs to pre-calculate lookup tables of atmospheric optical properties such as scattered path radiance or transmission for atmospheric states appropriate to the conditions observed at image acquisition. At runtime, the model inversion estimates the actual state from the radiance spectrum and interpolates within the lookup table to find the associated optical properties. This informs parametric approximations of atmospheric transport, such as the formulation by Vermote et al. (1997), permitting algebraic solutions for the remaining unknowns like surface reflectance. The sequential retrieval of atmospheric and surface properties, a process known as *atmospheric correction*, obtains a self-consistent but approximate explanation for the surface and atmosphere system.

The lookup table solution is adequate for many needs, but imposes several limitations. First, lookup tables can only model a few degrees of freedom in an atmospheric state due to the "curse of dimensionality;" the number of samples necessary to adequately represent the state space increases exponentially with the number of input variables. To increase the fidelity of grid samples in high dimensions, designers can leverage representative sampling or hyperparameter search strategies such as Snoek et al. (2012) within the state space, or space-filling sampling methods like Latin hypercube sampling (Stein, 1987) or lattice regression methods (Gupta et al., 2015). However, such techniques are restricted by prohibitive computation and storage requirements for highly dimensional state spaces, and incur increased risks of interpolation inaccuracy. Also, because the contents of precalculated lookup tables capture atmospheric optical properties independently from the surface, lookup-table-based approaches preclude strong coupling between the two. Speeding RTMs to the point at which they could run many times faster for each spectrum would obviate the lookup table compromise and enable more flexible, accurate, and statistically rigorous inversion algorithms such as the optimal estimation approach used in many atmospheric sounding missions (Thompson et al., 2018c; Rodgers, 2000).

Recent work suggested the use of nonparametric function approximators such as neural networks (Verrelst et al., 2016, 2017; Thompson et al., 2018a) or Gaussian processes (Martino et al., 2017) for this purpose. Investigators can train such models using prior runs of radiative transfer models over relevant ranges of surface and atmospheric conditions. After learning the underlying function with sufficient accuracy, the trained model could act as an instrument-specific RTM that would not have to solve the underlying differential equations. Alternative formulations such as Jamet et al. (2005) and Brajard et al. (2006) provide empirical validation of RTM assumptions by evaluating atmospheric, transmittance, and surface interactions captured in separate models, while other methods (e.g., Jamet et al., 2012; Kox et al., 2014; Loyola et al., 2018) permit retrieval of atmospheric or radiometric parameters based on models constructed using outputs generated by first-principle RTMs that span multiple wavelengths. However, to date, techniques designed to retrieve surface reflectance using learned RTM emulators have only been demonstrated on a small number of cases with limited surfaces and atmospheres (Verrelst et al., 2017; Martino et al., 2017; Brajard et al., 2006), and not across the VSWIR range with state vector flexibilities that would permit a functionally useful alternative for existing atmospheric correction routines (e.g., as a surrogate forward model). To our knowledge, this work represents the first demonstration on actual imaging spectroscopy data using nonparametric function approximation to emulate the RTM function $F(x\ \text{TS2}) \rightarrow y$ such that the RTM emulator is capable of acting as a forward model in an atmospheric correction procedure, thereby allowing us to retrieve surface reflectance over the entire VSWIR range for variable imaging conditions.

This study demonstrates an accurate neural network model deployed as part of an iterative model inversion, showing that emulation is a practical solution for operational atmospheric correction of imaging spectroscopy data. This opens new possible avenues of research, for both the inversion algorithm itself (to explore further expansions of the state vector beyond the traditional retrieved variables) and downstream analyses (to exploit the benefits of new retrieval methods that do not require lookup tables). We begin by describing the neural network architecture and RTM emulation methodology, including several novel advances: an *analytical decomposition* of the radiative transfer function $F(x)$ into quantities that are individually easier to model, *channelwise, monochromatic subnetworks* to simplify training and prediction, and *weight propagation* to account for correlation between adjacent channels and to reduce training time. We also describe an approach to partition the state space in a manner that guides each subnetwork to generate accurate predictions for states within the bounds of the state space. We evaluate our approach in a case study focusing on atmospheric correction for the PRISM imaging spectrometer, and demonstrate high-quality surface reflectance retrievals using the optimal estimation approach of Thompson et al. (2018c) equipped with our neural RTM as the forward model. The retrievals capture subtle atmospheric variability such as view dependence of Rayleigh scattering not typically handled in conventional atmospheric correction codes. Finally, we describe paths for future development of neural network RTM emulation technology.

## 2 Neural networks for radiative transfer modeling

Our goal is to construct a model that emulates a first-principle RTM using precalculated outputs generated by that RTM for a representative set of atmospheric, geometric, and surface states. More formally, we aim to model the RTM function $F(\boldsymbol{x}) \to \boldsymbol{y}$ that maps a set of $m$ distinct state parameters $\{p_j\}_{j=1}^m$ with values captured in a state vector $\boldsymbol{x} \in \mathbb{R}^m$ to a vector $\boldsymbol{y} \in \mathbb{R}^n$ of observed at-sensor radiances for $k$ channels centered at wavelengths $\{wl_1, \ldots, wl_k\}$. We use boldface notation to signify vectors and matrices, with all matrices in capital letters.

We exploit two features of the problem to simplify $F(\boldsymbol{x})$. First, we leverage the fact that the observed radiance at any given channel is fully specified by the observation geometry, atmospheric state, and the surface reflectance *in that channel*. In statistical terms, absent any prior distribution that couples neighboring wavelengths, the channelwise radiances become conditionally independent of each other given the atmosphere and observation geometry. This permits an exact decomposition of $F(\boldsymbol{x})$ into *monochromatic functions* $F(\boldsymbol{x}) = f_i(\boldsymbol{x})_{i=1}^k$, where each $f_i(\boldsymbol{x}) \to \boldsymbol{y}_i$ represents the RTM function for the channel centered at wavelength $wl_i$. Given this decomposition, we construct a neural RTM emulator using a set of $k$ channelwise subnetworks, where each subnetwork is trained to model a single $f_i$. Figure 1 shows the topology of one of the channelwise subnetworks in the neural RTM. A side benefit of this approach is that the partial derivatives of radiance channels with respect to their surface reflectances are independent of each other, which simplifies calculations of analytical Jacobians during iterative gradient descent inversions (Thompson et al., 2018c).

Second, we reduce the radiance spectrum analytically to the top-of-atmosphere reflectance, written $\boldsymbol{\rho}_{\mathrm{obs}}$, and solar illumination components. The top of atmosphere reflectance is defined as $\boldsymbol{\rho}_{\mathrm{obs}} = \boldsymbol{y}\pi/\phi_o\boldsymbol{e}_o$, where $\phi_o$ is the cosine of the solar zenith angle and $\boldsymbol{e}_o$ the extraterrestrial solar irradiance. $\boldsymbol{\rho}_{\mathrm{obs}}$ is normalized for solar illumination and, absent extreme glint, resides conveniently in the $[0, 1]$ interval, making it an easier target for function approximation. For any given observing geometry, the known values of $\phi_o$ and $\boldsymbol{e}_o$ can be used to infer the corresponding radiances.

Constructing a robust neural RTM emulator from precomputed RTM outputs faces two fundamental modeling challenges. First, the precomputed RTM outputs must provide sufficient coverage of the state space to represent the distribution of spectral responses in each channel. Second, the subnetworks must accurately predict RTM outputs for intermediate state parameter values within the bounds of the precomputed state space for all channels. Intuitively, modeling channels whose RTM outputs are stable in state is easier than modeling channels whose spectral responses vary substantially with respect to small changes in state. For instance, varying concentrations of atmospheric water vapor produce complex, nonlinear behavior for water absorption

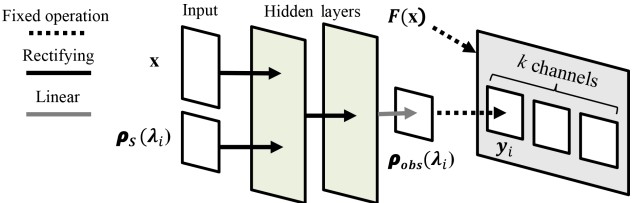

**Figure 1.** Illustration of a single subnetwork in the neural RTM emulator. Each subnetwork predicts the top-of-atmosphere reflectance $\boldsymbol{\rho}_{\mathrm{obs}}(\lambda_i)$ for a single channel centered at wavelength $\lambda_i$ provided state parameters $\boldsymbol{x}$ and surface reflectance $\boldsymbol{\rho}_{\mathbf{s}}(\lambda_i)$. Collecting the predictions generated by $k$ subnetworks, each modeling distinct channels, and converting those predictions from $\boldsymbol{\rho}_{\mathrm{obs}}$ to radiance emulates the RTM function $F(\boldsymbol{x}) \to \boldsymbol{y}$ for the selected channels.

bands, while other wavelengths are largely unaffected. Accurately modeling unstable channels may require generating additional RTM outputs at increased sampling density to highlight distinct responses that are poorly represented in the existing precomputed outputs, and additional computational resources to fine-tune the subnetwork to capture those distinctions may also be in order.

To ensure each subnetwork reliably models its corresponding channel, we measure prediction accuracy on a test set of precalculated RTM outputs excluded from the training process. In our initial experiments, we performed both $k$-fold cross-validation and $k$-fold bootstrap sampling, but after we observed that the main sources of variability in the state space emerged from interactions among a small number of state parameter values, we concluded that randomized sampling of the state space without an informed sample stratification yields optimistic or inconsistent estimates of test accuracy and/or convergence time in cross-validation. Ultimately, we concluded that validation using a fixed and bounded subset of the state parameter values would provide a more informative assessment of model performance. Using a bounded subset also permitted direct comparison to lookup-table-based approaches, as they require upper and lower bounds on each variable to generate intermediate values via interpolation. We describe this approach in more detail later in this work.

We can also improve model accuracy and reduce computational demands by exploiting characteristics of the state space in tandem with RTM modeling assumptions. One means we use to achieve this is through a process of *weight propagation*. Rather than initializing the weights for each subnetwork from scratch, we use the converged weights of the subnetwork modeling the previous channel to initialize the weights for the subnetwork modeling the current channel, which are then fine-tuned to estimate the spectral responses for the current channel to sufficiently low test error, as before. In comparison to training each subnetwork from scratch, using weight propagation often yields a substantial reduction in training time, along with improved accuracy for channels

whose RTM outputs are relatively stable with respect to the state space. An additional side benefit is that weight propagation provides an approximate means to account for channel-wise coupling for instruments whose spectral response functions for neighboring channels partially overlap. In practice, requiring the first subnetwork to converge to a lower test error than the subsequent networks can help ensure that the propagated weights will be informative for subsequent channels. We can similarly apply different stopping criteria for subnetworks initialized with weight propagation representing poorly correlated neighboring channels to increase the likelihood that training converges to the appropriate channelwise responses.

Algorithm 1 describes the procedure to train the neural RTM emulator provided $n$ samples from the $m$-dimensional state space and their corresponding $\boldsymbol{\rho}_{\mathrm{obs}}$ outputs, each spanning $k$ channels. The output of the algorithm is a trained neural RTM that takes a state vector $\boldsymbol{x}$ of $m$ parameters as input and outputs a $k$-dimensional prediction vector. The output radiance vector $\boldsymbol{y}$ is the concatenated output produced by the trained subnetworks $\{f_i\}_{i=1}^{k}$ and converted to radiance with respect to $\phi_o$ and $\boldsymbol{e}_o$ for each of the $k$ channels.

Our goal is to train each subnetwork $f_i$ to generate accurate predictions for states explicitly included in $\mathbf{X}$ and also (more importantly) for *intermediate* states not explicitly included in $\mathbf{X}$ but within the bounds of the state space. To achieve this, we use a sampling strategy that partitions the state space into training and test sets in a manner that helps optimize each subnetwork to accurately predict intermediate states. We first partition $\mathbf{X}$ and $\mathbf{Y}$ into disjoint training $(\mathbf{X}, \mathbf{Y})^{\mathrm{tr}}$ and test $(\mathbf{X}, \mathbf{Y})^{\mathrm{te}}$ sets such that $\mathbf{X}^{\mathrm{tr}}$ contains all state vectors containing the boundary values $\{\min(p_j), \max(p_j)\}$ of all state parameters. This partitioning ensures the training set contains the convex hull of the Euclidean subspace of $\mathbb{R}^m$ defined by the state parameters and also that all test states in $\mathbf{X}^{\mathrm{te}}$ represent intermediate states within the hull. To capture the internal structure of the state space within the hull, the training set should also contain one or more intermediate state vectors for each $p_j$ satisfying $\min(p_j) < \boldsymbol{x}_j^{\mathrm{tr}} < \max(p_j)$. Given the training and test partitions, we train each subnetwork to model $f_i$ by minimizing the $L^2$-regularized mean-squared error (MSE) between the predicted and the observed values of the $n^{\mathrm{tr}}$ training samples $(\mathbf{X}, \boldsymbol{y}_i)^{\mathrm{tr}}$ representing the $\boldsymbol{\rho}_{\mathrm{obs}}$ responses for the $i$th channel.

We use a feed-forward architecture with two hidden layers, rectifying linear activation functions in the hidden layers, which have been shown to be more robust than the conventional sigmoid/tanh CE2 activations used in traditional neural networks (Nair and Hinton, 2010), and a linear activation in the output layer for each subnetwork. We use the method proposed by Glorot and Bengio (2010) to initialize subnetwork weights when necessary, and use the widely used error back propagation algorithm (Werbos, 1982) with adaptive moment estimation (Kingma and Ba, 2014) to optimize the weights via gradient descent. We train each subnetwork until the er-

**Table 1.** State parameters values used in libRadtran model runs to generate $\boldsymbol{\rho}_{\mathrm{obs}}$ spectra to train and validate the neural RTM. State vectors containing the median value of each auxiliary parameter (indicated by bold text) are held out for testing, while the remaining state vectors are used for training the channelwise subnetworks.

| State parameter | State values |
| --- | --- |
| Solar azimuth ($\phi_{\mathrm{r}}$) | $0, \frac{\pi}{8}, \ldots, \frac{\boldsymbol{\pi}}{\mathbf{2}}, \ldots \frac{7\pi}{8}, \pi$ |
| Observer zenith angle ($\cos(\theta_{\mathrm{v}})$) | 0.94, 0.95, 0.96, **0.97**, 0.98, 0.99, 1.0 |
| Aerosol optical depth ($\tau$) | 0.05, 0.1, **0.2**, 0.3 |
| Water vapor ($\mathrm{H_2O}$) | 0, 0.5, 1.0, **1.5**, 2.0, 2.5 |
| Surface reflectance ($\rho_{\mathrm{s}}$) | 0.05, 0.1, 0.25, 0.5, 1.0 |

ror converges to within 0.1 % mean absolute error (MAE), or we reach the maximum number of epochs $n_{\mathrm{epoch}}$. This level of accuracy is sufficient to make the approximation error a smaller contributor to total uncertainty than other unknowns in the measurement system. For example, it is generally a similar magnitude to relative calibration error of different focal plane array elements, which can vary slightly due to drift between calibrations (Thompson et al., 2018a).

## 3 Neural RTM emulation for PRISM

We define a case study demonstrating the capabilities of our RTM emulator for atmospheric correction on data acquired by the PRISM imaging spectrometer (Mouroulis et al., 2008, 2014). PRISM is a push-broom design and observes a cross-track angular field of view spanning approximately 30°, and is designed to observe coastal ocean environments in the 350–1050 nm spectral range at approximately 3 nm spectral sampling. The instrument was mounted onboard a high-altitude ER-2 aircraft which overflew Santa Monica, USA, in October 2015 at 20 km above sea level (Thompson et al., 2018b; Trinh et al., 2017). At this altitude, the instrument measured the scene through nearly all of Earth's atmospheric scattering and absorption, providing a challenging test case with relevance to future orbital instruments.

Our state space consists of the surface reflectance $\boldsymbol{\rho}_{\mathrm{s}}$, represented by a single free parameter per instrument channel, along with $m = 4$ state parameters captured in state vector $\boldsymbol{x}$ representing a concise but representative suite of parameters used operational settings. These include the atmospheric aerosol optical depth at the surface, $\tau$; the atmospheric water vapor content of the column in grams per square centimeter, $\mathrm{H_2O}$ TS3; the cosine of the observer zenith angle, $\cos(\theta_{\mathrm{v}})$; and the relative azimuth angle between the observer and the Sun, written $\phi_{\mathrm{r}}$. Each of these free parameters varies independently for every spectrum in a given flight line. Naturally, alternative parameterizations are possible, including mixture models, continuum-absorption models, and others. However,

Please note the remarks at the end of the manuscript.

---

**Algorithm 1** Neural RTM Training

---

**Input:** $n \times m$ matrix $\mathbf{X}$ of $n$ state vectors, each representing $m$ parameters; $n \times k$ matrix $\mathbf{Y}$ of $k$-dimensional $\boldsymbol{\rho}_{obs}$ spectra associated with each state vector at wavelengths $\lambda = \{\lambda_i\}_{i=1}^k$; binary mask of training indices $\mathbf{v} = \{v_i\}_i^n$, $v_i \in \{0,1\}$; number of neural network layers $\ell$; number of hidden neurons in each layer $\{h_l\}_{l=1}^\ell$; convergence tolerence $tol$; maximum number of training epochs $n_{epoch}$

**Output:** Neural RTM model $F(\mathbf{x}) \to \mathbf{y}$ consisting of $k$ trained neural network regressors $f_i$ each mapping $m$ dimensional state vector $\mathbf{x}$ to corresponding $\boldsymbol{\rho}_{obs}$ response $\mathbf{y}_i$ at wavelength $\lambda_i$

  **for** $i = 1$ **to** $k$ **do**

    Let $\mathbf{y}_i = \mathbf{Y}_{\cdot,i}$ be the $\boldsymbol{\rho}_{obs}$ responses at wavelength $\lambda_i$ associated with the $n$ state vectors.

    Partition $(\mathbf{X}, \mathbf{y}_i)$ into $\{(\mathbf{X}, \mathbf{y}_i)^{tr}, (\mathbf{X}, \mathbf{y}_i)^{te}\}$ using training indices $\mathbf{v}$.

    Let $f_i$ be an $L$-layer neural network model with set of weight matrices $W_i = \{\mathbf{W}_l\}_{l=1}^L$ and corresponding bias vectors $b_i = \{\mathbf{b}_l\}_{l=1}^L$.

    **if** $i = 1$ **then**

      Initialize new model $r_1$ for first channel by populating $W_1, b_1$ with random values (via (Glorot and Bengio, 2010)).

    **else**

      Propagate weight matrices and bias vectors from previous model $r_{i-1}$ to current model $f_i$ via $W_i = W_{i-1}, b_i = b_{i-1}$.

    **end if**

    **for** $e = 1$ **to** $n_{epoch}$ **do**

      Train $f_i$ to minimize $\boldsymbol{\rho}_{obs}$ prediction error for channel centered at wavelength $\lambda_i$ based on training set $(\mathbf{X}, \mathbf{y}_i)^{tr}$.

      Compute average error $e_{test}$ applying $f_i$ to test set $(\mathbf{X}, \mathbf{y}_i)^{te}$.

      **if** $e_{test}$ has converged to tolerance $tol$ **then**

        **return** Trained model $f_i$

      **end if**

    **end for**

  **end for**

  **return** Trained neural RTM $F(\mathbf{x}) \to \mathbf{y} = \{f_i(\mathbf{x}) \to \mathbf{y}_i\}_{i=1}^k$

---

these could be mapped to our representation without loss of generality.

We identified a set of values for each state parameter that covered the anticipated range of conditions that could occur during the flight campaign, and provided those values in Table 1. We generated RTM outputs using the libRadtran radiative transfer code (Emde et al., 2016; Mayer and Kylling, 2005) with midlatitude summer atmosphere appropriate for the PRISM flight line we considered in this study. We provide the template libRadtran config file in the Supplement (prm20151026t173148_libradtran_config) (Kurucz, 1994; Buehler et al., 2009; Bodhaine et al., 1999). Generating $\boldsymbol{\rho}_{obs}$ spectra for every combination of state parameter values yielded $n = 9072$ total $\boldsymbol{\rho}_{obs}$ output spectra, each of $k = 7101$ dimensions spanning the range of the PRISM instrument wavelengths with 0.1 nm spacing. Our test data consist of the set of all state vectors containing the median value of each state parameter (shown in bold text in Table 1) and the $\boldsymbol{\rho}_{obs}$ spectra associated with those states. The remaining states and their corresponding $\boldsymbol{\rho}_{obs}$ spectra form our training set.

Figure 3 depicts the changes in the $\boldsymbol{\rho}_{obs}$ spectra with respect to parameters $\phi_r$ (panel a), $\cos(\theta_v)$ (panel b), $\tau$ (panel c), and $H_2O$ (panel d), while holding the other parameters fixed at their median values. Unsurprisingly, the most visibly dramatic changes occur as absorption features appear with increased $H_2O$ vapor concentrations. Of the remaining auxiliary parameters, only aerosol optical depth has an observable effect on spectral shape across the visible and near-infrared wavelengths. Changes for varying $\phi_r$ and $\theta_v$ are comparatively small and predominantly observable in the visible range.

For this case study, we focused on modeling $F(\boldsymbol{x})$ based on libRadtran outputs resampled to the PRISM instrument channels. This dramatically reduces the computation required to construct the neural RTM, as we only needed to train a total of 245 subnetworks representing each of the PRISM channels with 2.83 nm spacing, rather than the 7101 channels at 0.1 nm spacing generated by libRadtran. As a consequence of convolving the libRadtran spectra to the lower-resolution PRISM spectral response function (SRF) at each wavelength, the $\boldsymbol{\rho}_{obs}$ values are no longer strictly monochromatic, but the instrument channels are well-separated so that channelwise coupling should not be a significant issue. We plan to construct a more general neural RTM that generates $\boldsymbol{\rho}_{obs}$ predictions at 0.1 nm spacing, which are then convolved to the spectral response function associated with a particular sensor.

We observed experimentally that subnetworks consisting of two hidden layers with 50 units each and a training cycle of at most 500 epochs (where one epoch consists of a full pass of gradient updates over the training set) with batch sizes ranging from 100 to 200 training samples was sufficient for each subnetwork to converge to our error re-

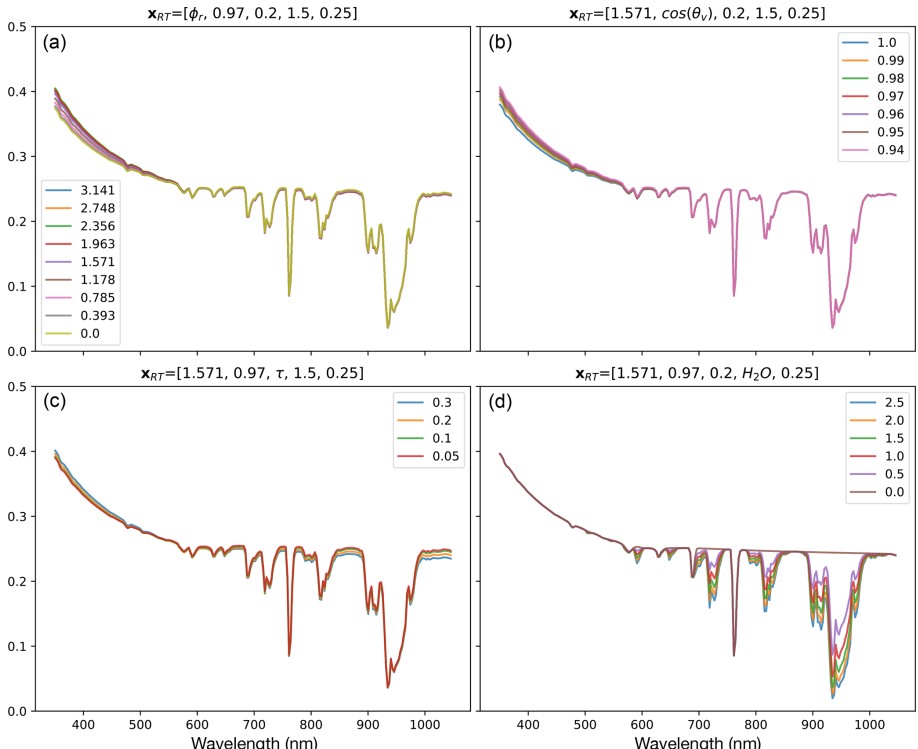

**Figure 3.** $\rho_{\text{obs}}$ spectra for $\rho_{\text{s}} = 0.25$ spanning the range of the $\phi_{\text{r}}$ **(a)**, $\cos(\theta_{\text{v}})$ **(b)**, $\tau$ **(c)**, and $H_2O$ **(d)** parameters with respect to the validation grid values.

quirements for the state space parametrized by values in Table 1. Notably, single-layer networks were often insufficient to model channels whose $\rho_{\text{obs}}$ responses changed in a highly variable and/or nonlinear manner with small changes in the state parameters (e.g., the water absorption bands). We set the initial learning rate to 0.001 with the following adaptive moment estimation parameters $\{\beta_1 = 0.9, \beta_2 = 0.999, \epsilon = 10^{-10}\}$ and set the $L^2$ regularization penalty term $\alpha$ to $10^{-4}$ for each subnetwork. A longer training cycle or additional hidden units can be used to match the RTM output more precisely and would likely be necessary to model more complex state parameter spaces.

As a baseline comparison, we used the channelwise training samples $((\mathbf{X}, \boldsymbol{y}_i)^{\text{tr}}$ in Algorithm 1) to train a least-squares linear regressor on the $\rho_{\text{obs}}$ responses for each channel, and applied each regressor to generate predictions on the associated test samples $(((\mathbf{X}, \boldsymbol{y}_i)^{\text{te}}$ in Algorithm 1)). The channelwise test errors using the linear regressors provide an approximate upper bound of the error that would be incurred using piecewise/locally linear interpolation to infer $\rho_{\text{obs}}$ responses for intermediate states based on lookup tables. Figure 4 compares the $\rho_{\text{obs}}$ test prediction error using the channelwise linear regressors (panel a, black line) to the error produced by the channelwise subnetworks trained from scratch (NN, blue line) versus channelwise subnetworks initialized with weight propagation (NN$_{\text{WP}}$, red line). The channelwise sub-

networks yield an order of magnitude reduction in prediction error on all channels in comparison to the linear regressors, and demonstrates potentially significant issues with lookup-table-based approaches. Weight propagation provides an average reduction of 64 % in channelwise error, but also yields systematically higher errors in the $H_2O$ absorption range between 890 and 1000 nm where the $\rho_{\text{obs}}$ responses vary rapidly for adjacent channels.

While it is unsurprising that the $H_2O$ absorption wavelengths are challenging to model, the fact that the two weight initialization schemes yield distinct error distributions for those wavelengths suggests model convergence issues. Figure 5 compares the number of epochs – where one epoch consists of a single pass of weight updates over all samples in the training set – required to converge for channelwise subnetworks trained from scratch versus subnetworks initialized with weight propagation. Over the set of all PRISM instrument channels, weight propagation permits convergence in ≈ 70 % fewer epochs over subnetworks not leveraging weight propagation. In terms of raw compute time, our scikit-learn (Pedregosa et al., 2011) implementation requires 2–3 min to train a single monochromatic subnetwork from scratch on a single commercial processor core, while subnetworks initialized with weight propagation typically require less than 30 s to converge. However, we note that the channelwise subnetworks trained with weight propagation converge as quickly

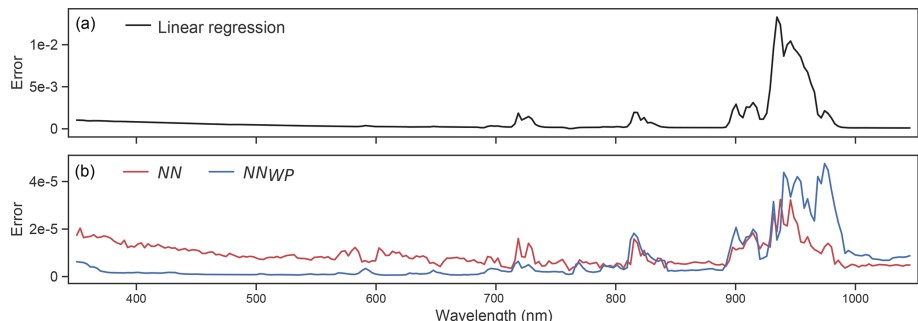

**Figure 4. (a)** $\rho_{\mathrm{obs}}$ test prediction error per channel using channelwise linear regressors (black line). **(b)** Neural network test prediction error per channel using subnetworks trained from scratch ($NN$, red line) versus subnetworks trained with weight propagation (NN$_{\mathrm{WP}}$, blue line).

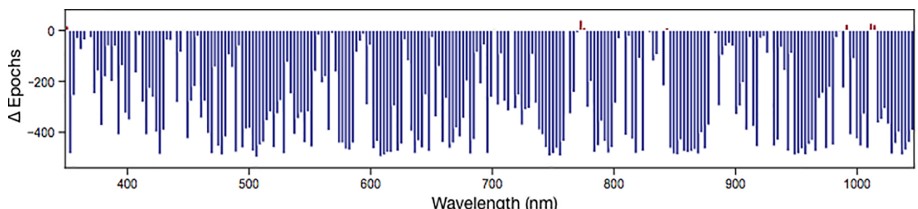

**Figure 5.** Difference in training epochs to converge to 0.1 % validation error. Negative values (blue bars) show channelwise subnetworks that converged faster using weight propagation, while positive values (red bars) indicate channels whose subnetworks converged more quickly when trained from scratch.

in the 925–975 nm range – where their most significant prediction errors occur – as in the remaining channels.

Investigating further, we measured the average root-mean-square error (RMSE) on the test set with respect to the pairwise interactions between $\rho_{\mathrm{s}}$ and the four state parameters, and show the resulting error surfaces in Fig. 6. Relatively small errors for the majority of the parameter space indicate that the $\rho_{\mathrm{obs}}$ spectra vary smoothly with respect to most state parameter values, with the most significant variability emerging from a small range of values in the $\rho_{\mathrm{s}} \in [0.4, 0.8]$ and $H_2O \in [1.0, 2.0]$ regions of the state space. The relatively high error in this regime is consistent with our earlier observation that small changes in the atmospheric water vapor parameter yield considerably different $\rho_{\mathrm{obs}}$ spectra, as shown in Fig. 3, and the comparatively high prediction errors for the $H_2O$ absorption bands shown in Fig. 4.

## 4 Atmospheric correction with the neural RTM emulator

We now evaluate the neural RTM emulator in the context of a surface–atmosphere retrieval problem, retrieving surface reflectance for comparison to known surface materials. To that end, we fused the optimal estimation (OE) formalism of Rodgers (2000), following the specific approach of Thompson et al. (2018c) for application to imaging spectroscopy. The OE method estimates the atmosphere and surface state vector by an iterative least-squares optimization of the for-

ward model's match to the measured radiances. Cost terms related to observation error and prior probabilities of state vector elements ensure rigorous propagation of uncertainties in the retrieval.

Continuing our case study, we begin by computing radiometric calibration factors for the PRISM flight line via vicarious calibration. This procedure, similar to standard practice calibration for imaging spectrometers (Thompson et al., 2018a), projects the residual error in retrieved surface reflectance back into radiance space where it becomes a multiplicative correction factor applied independently to each channel. We generate a "standardized" surface reflectance target by performing a first-principle retrieval for a beach sand radiance spectrum manually selected from the target PRISM image. We smooth the resulting surface reflectance spectrum to suppress significant atmospheric features, and use the smoothed spectrum to generate radiometric correction factors appropriate to our flight line. Applying the resulting factors fine tunes the calibration for optimal results and suppresses residual errors caused by uncertainty in spectral response or RTM inaccuracy. For reference, the beach sand radiance spectrum and the resulting smoothed surface reflectance spectrum are shown in Fig. 7.

We applied the atmospheric correction procedure to a set of radiance spectra from the PRISM flight line representing a diverse range of surface materials including grass, rooftop materials, soil, and seafoam. Figure 8 shows a successful retrieval result for a radiance spectrum representing grass on a golf course fairway. The inversion (orange line) perfectly

**www.atmos-meas-tech.net/12/1/2019/** **Atmos. Meas. Tech., 12, 1–12, 2019**

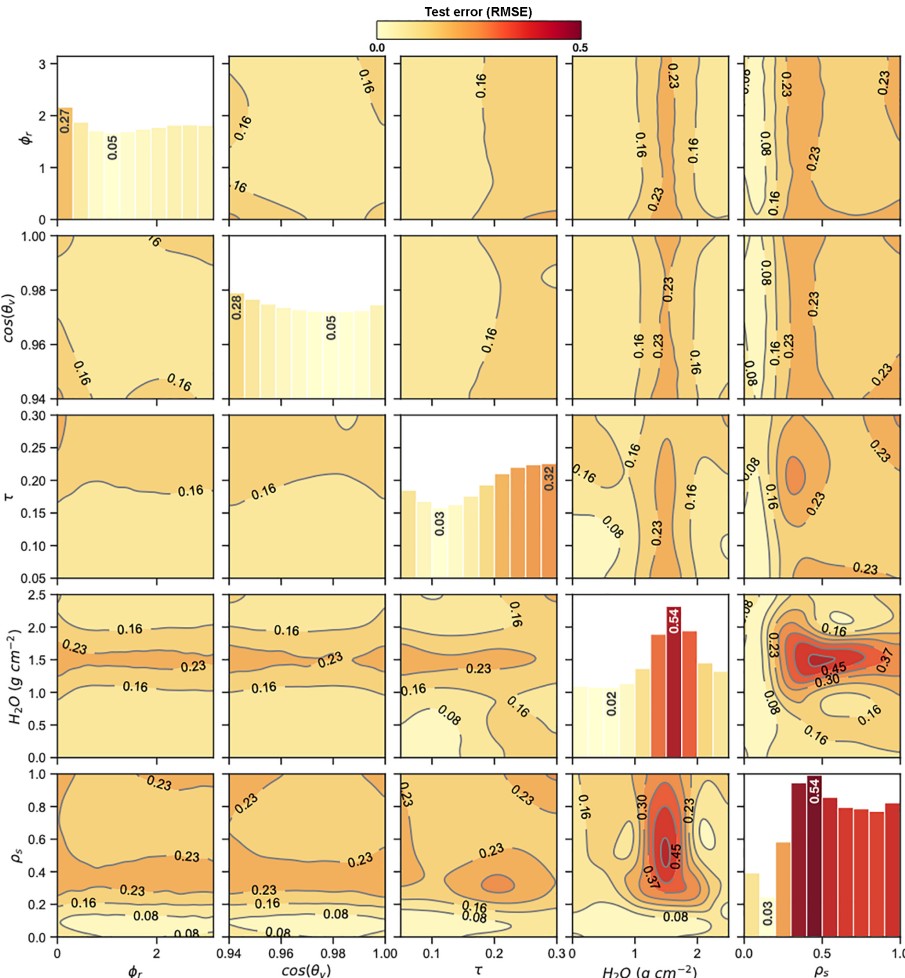

**Figure 6.** Pairwise contour plots showing the $\rho_{obs}$ test prediction error (RMSE) surfaces with respect to the state parameter values specified in Table 1. Contour labels on the off-diagonal subplots give the error levels associated with each contour. Diagonal subplots show the average RMSE in 10 uniformly spaced bins spanning the ($x$ axis) range of each parameter. Vertical labels on the diagonal subplots indicate the minimum and maximum error values for each parameter and their corresponding bins.

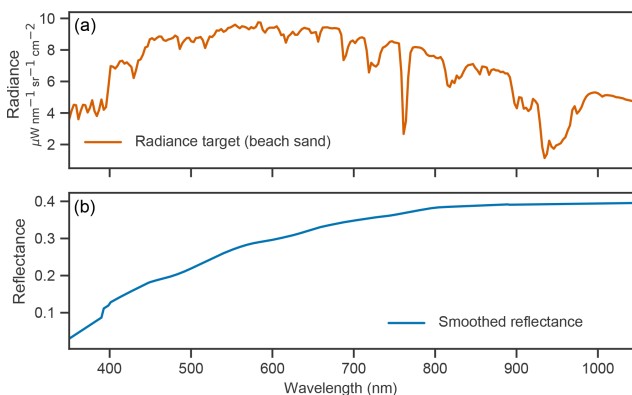

**Figure 7.** PRISM radiance spectrum **(a)** and the resulting smoothed reflectance **(b)** spectrum for the beach sand target used in the vicarious calibration procedure.

matches the measured radiance (black dashed line) in Fig. 8a. In Fig. 8b, the estimated surface reflectance (blue line) is an extremely smooth and faithful estimate of a dark vegetation spectrum. Figure 9 shows additional radiance spectra (panel a) and their corresponding surface reflectance retrievals (panel b). The high-quality surface reflectance estimates – evidenced by the lack of residual bumps caused by atmospheric absorption and the flat, low surface reflectance profiles in the aerosol-dominated interval from 400 to 450 nm – provide additional confidence in the network's value for atmospheric correction. Our neural RTM emulator runs in less than 5 ms per PRISM spectrum (about 0.02 ms per channel). This represents a reduction of several orders of magnitude in runtime in comparison to analogous first-principle RTMs (i.e., monochromatic RTMs that solve the coupled scattering–absorption problem in a computationally exact manner, such as DISORT), which typically required over

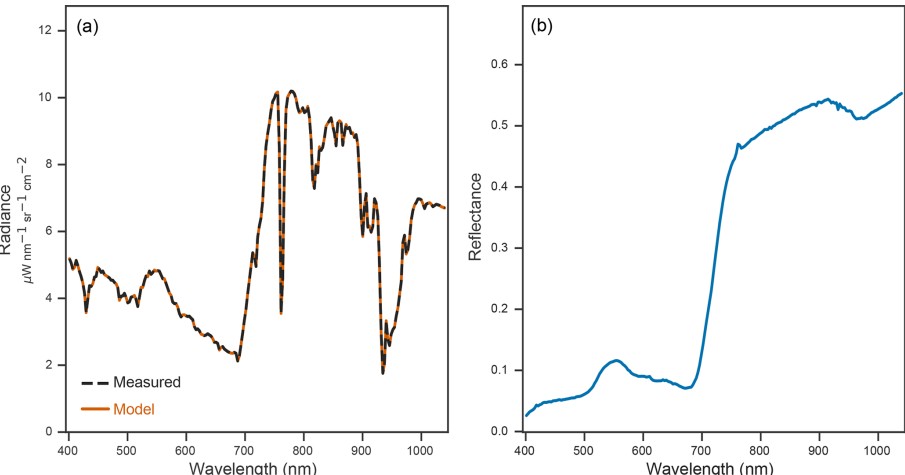

**Figure 8.** Example surface reflectance retrieval for a PRISM vegetation spectrum. Panel **(a)** shows the measured (orange CE3) versus predicted (black) radiance spectra. Panel **(b)** shows the retrieved surface reflectance spectrum (black).

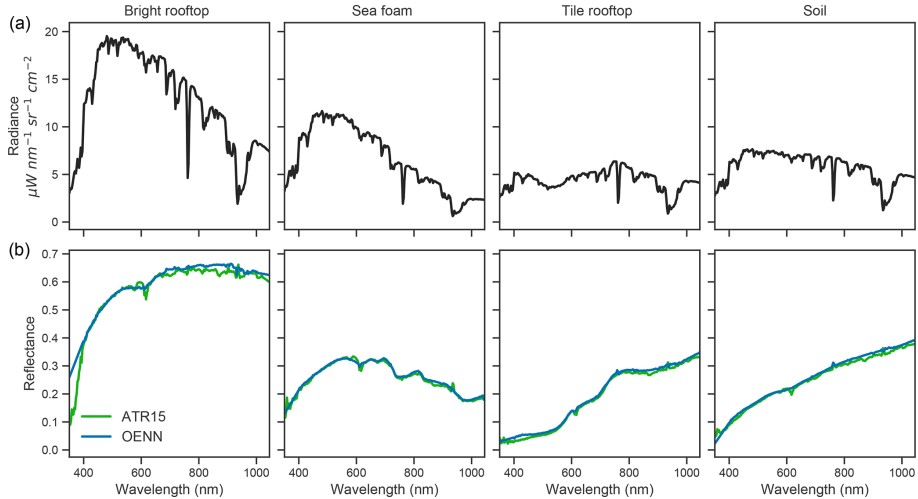

**Figure 9.** Selected radiance spectra **(a)** and corresponding surface reflectance retrievals **(b)** using the ATREM-based atmospheric correction approach of Thompson et al. (2015) (ATR15, green spectra) versus optimal estimation equipped with our neural network RTM emulator as the forward model (OENN, blue spectra).

10 min to generate a spectrum at 0.1 nm spacing (about 0.15 s per channel).

## 5 Conclusions

Neural network RTM emulation offers a path to reduce both interpolation inaccuracy and simultaneously runtime by several orders of magnitude. A well-parametrized neural RTM is capable of modeling state parameter spaces with significantly higher accuracy than conventional lookup-table-based approaches. Such high-capacity statistical models have potential for modeling state parameter spaces with much higher dimensionality than current methods.

The computational and theoretical advantages provided by fast and accurate RTM emulators are particularly useful for iterative approaches that must recalculate the entire forward model many times for each spectrum. Equipping iterative formalisms such as optimal estimation with the neural RTM forward model also enables new retrieval approaches that jointly estimate surface and atmospheric parameters. Joint retrieval of surface and atmospheric parameters carries several advantages. It becomes possible to estimate arbitrary parameters of the atmospheric state simply by adjusting the RTM dynamically during the fitting process. A joint retrieval can represent strong coupling between surface and atmosphere, including bidirectional reflectance distribution function (BRDF) effects, and obviates parametric approximations. The ability to model strong coupling is particu-

larly important for conditions with off-nadir views or haze. Finally, a combined model enables a rigorous, unified, and quantitative treatment of uncertainty, respecting uncertainties in all measurement processes and modeled variables and propagating posterior uncertainties for downstream analysis.

Our results also demonstrate the advantages of informed sampling of the state space. Finer grid sampling in rapidly varying regions of the state space is advantageous to capture complex and often nonlinear interactions among state parameters, while coarse sampling is beneficial in regions of the state space that vary smoothly to reduce redundancy and computational overhead. Uninformed sampling of the state space may not only lead to inaccurate models, but can also yield overly optimistic or inconsistent results when measuring test accuracy or convergence time during cross-validation. For example, as Figs. 3 and 6 indicate, much of the state space is relatively smooth. Traditional cross-validation strategies that randomly partition the state space into training and test sets will indicate the subnetworks generalize well due to sampling bias in regions of the state space that are easy to model. Sample stratification approaches during cross-validation can help to ensure each subnetwork accurately captures the parameters that are more difficult to model. However, an informed sampling of the state space would not only eliminate the need for sample stratification during cross-validation, but would also ultimately yield more accurate models with reduced computational overhead.

Future work will train a more "universal" neural RTM designed to generate $\rho_{\mathrm{obs}}$ predictions at much spectral resolution CE4 for a comprehensive set of states, and are also investigating Bayesian optimization or smart sampling approaches (e.g., Loyola R et al., 2016) that may provide an informed sampling of the state parameter space. We also aim to reduce approximation error still further, in order to keep the fractional contribution small for very dark and/or noisy targets, and are considering re-parameterizing the model to retrieve additional aerosol optical properties.

*Code availability.* The python code used to train and apply the neural RTM for optimal-estimation-based atmospheric correction is available at the following URL: https://github.com/dsmbgu8/isofit/ (last access: TS4). TS5

*Supplement.* The supplement related to this article is available online at: https://doi.org/10.5194/amt-12-1-2019-supplement.

*Author contributions.* Authors BDB and DRT conceived, implemented, and described the methodology and experiments described in this paper, and are the primary contributors to this work. SD contributed model analysis and RTM data collection during a Caltech Summer Undergraduate Research Fellowship (SURF) internship at JPL in summer 2017. ME and ROG provided expertise in the theory and application of imaging spectroscopy and atmospheric correction methods. VN provided expertise in radiative transfer modeling along with software and guidance for future efforts. TM contributed analysis and generated RTM outputs during a Caltech SURF internship at JPL in summer 2018. MP provided support and guidance for TM. CE5

*Competing interests.* The authors declare that they have no conflict of interest.

*Acknowledgements.* We thank Sven Geier, Scott Nolte, and the AVIRIS-NG instrument and Science Data System teams for assistance in calibration and operations. Particular thanks go to Charles E. Miller and Michael Turmon whose feedback significantly improved the paper. We are also thankful for the counsel of colleagues including Phil Townsend, Phil Dennison, Dar Roberts, Steven Adler-Golden, Alexander Berk, Steven Massie, and Bruce Kindel. We acknowledge the support of the NASA Earth Science Division for the AVIRIS-NG instrument and the data analysis program "Utilization of Airborne Visible/Infrared Imaging Spectrometer Next Generation Data from an Airborne Campaign in India" NNH16ZDA001N-AVRSNG, managed by Woody Turner, for its support of the algorithm development. We are also thankful for the support of the Jet Propulsion Laboratory Research and Technology Development Program, the NASA Center Innovation Fund managed in conjunction with the Jet Propulsion Laboratory Office of the Chief Scientist and Technologist, and the Caltech SURF program. A portion of this research took place at the Jet Propulsion Laboratory, California Institute of Technology. US government support is acknowledged.

*Review statement.* This paper was edited by Lars Hoffmann and reviewed by two anonymous referees.

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

## Remarks from the language copy-editor

## Remarks from the typesetter