# Peer review of "Neural Network Radiative Transfer for Imaging Spectroscopy"

_Atmospheric Measurement Techniques, 2018_

## Referee Comment (RC1) · Anonymous Referee #1 · 2 Feb 2019

This paper is a compact presentation of a novel method of the emulation and use of radiative transfer models in compensating for the effects of illumination and the atmosphere to retrieve surface reflectance from measured radiances collected by an imaging spectrometer. It is quite dense and while it presents an adequate description, the results are just limited examples of the accuracy and a demonstration of its use.

It would be nice to see an expanded treatment of the first part of the article, the development of the forward emulation model, which is the main contribution. Section 2 presents the model with many details presented in a dense manner. It would be nice to expand and include more explicit equations for the various steps involved. More discussion on the selection and justification of the approach would also provide more insight. In Section 3, the rationale behind limited the parameters of Table 1 would be

good to see.

In Figure 1, is the rho in the middle of the figure supposed to rho_obs? If so please label as such. If not, how does it relate to terms defined in the text?

Figure 5 should have a color legend to better interpret the quantitative results. Also, why show mean squared error? Why not root mean square which can be better interpreted? Actually, is this the mean error across the wavelengths? That would mask errors that may be concentrated in the water vapor region. Please clarify.

What was the source for the smoothed reflectance shown in Figure 6? Field spectrometer measurements? A library? Did you also use a dark target for the vicarious procedure?

This work presents an exciting development in the operational use of imaging spectrometers and deserves a more comprehensive presentation. Perhaps this could be the subject of future articles.

---

## Referee Comment (RC2) · Anonymous Referee #2 · 22 Feb 2019

The paper presents a neural network developed to calculate radiative transfer for the solar spectral region in clear sky. The model is applied to retrieve the surface spectral reflectance function from PRISM (airborne imaging spectrometer) data. As examples the retrieved surface reflectance spectra for 5 different surface types are shown to demonstrate that the method works well. The neural network method highly accelerates the atmospheric correction methodology, thus it is certainly an interesting approach which might become standard since the data to be processed increases rapidly with improvements in spatial and spectral resolution of sensors. The topic of the paper fits well in the scope of AMT, however, it needs to be revised, because the methodology needs to be described more precisely. Further the authors need to point, what makes their approach novel compared to other neural networks based approaches. I

recommend publication in AMT after these revisions.

Major comments:

1. The application of neural networks for remote sensing has become rather common in the last decade (besides the mentioned references e.g. Kox et al. 2014, Loyola et al. 2016, Eferemeko et al 2017, Strandgren et al. 2017, Loyola et. al 2018). The authors say that they for the first time apply a neural network based algorithm to imaging spectroscopy. I am not sure whether this is correct. At least, the operational cloud retrieval algorithm for TROPOMI, a spaceborne imaging spectrometer, also applies a neural network RTM (Loyola et al., 2018). So in my opinion the authors should point out, what makes their approach novel and what is the difference to other similar approaches.

If I understood correctly, the physical knowledge that is incorporated is the "analytical decomposition of the radiance spectrum into quantities that are individually easier to model". Later I find, that this means to use reflectances ("pi\*y/phi0\*E0" with y-radiance, phi0 - solar zenith angle and E0 -extraterrestrial irradiance; apart from E0 very uncommon nomenclature) instead of radiance. This is a very common approach, also in look-up-table based retrieval algorithms and not a new idea. I think that all publications mentioned below use reflectances and not radiances.

The second "novel" idea that is mentioned is the channel-wise training. This is as far as I know already applied in Kox et al. 2014. The only new idea, that I have not seen so far, is the channelwise weight propagation, which makes sense to me.

2. The setup of the neural network needs a more comprehensive description: - Most symbols in Figure 1 are not explained - It is not clear to me, whether the "monochromatic subnetworks" (p4, I.4) are really monochromatic (for 1 wavelength) or for one channel (convolved with the spectral response function). E.g. on p5, I4 it reads "To ensure each subnetwork reliably models its corresponding channel ..." - Explain settings: "feed-forward architecture with two hidden layers of 50 units each ... Why is the neural network set up as shown in Figure 1? Why are these particular settings chosen for this particular application? - Explain terms "rectifying" and "early stopping"

3. The radiative transfer simulations for training and validation were performed using the libRadtran package, which is a comprehensive package including various parameterizations, atmospheric setups, aerosol types etc. The setup of libRadtran needs to be specified, i.e. which atmosphere is used, which absorption parameterization, how is the aerosol defined, which radiative transfer solver is used etc. The symbols in Table 1 (input parameters) are also not defined.

4. Figure 3: How is the linear regression done? Linear interpolation in a lookup-table for the parameters listed in Table 1? This is not explained in the text.

5. Why are only 5 examples of retrieved surface reflectivity spectra shown. Since the method is so fast, it could be easily be applied on a full image. The difference between the traditional atmospheric correction approach and the new neural network approach should be assessed in a statistical sense.

Minor comments:

- Figure 5: I suggest to create a separate figure for the histograms shown in the diagonal with proper y-axis labels.

- It is confusing that the term reflectance is used for the top-of-atmosphere reflectance and the surface-reflectance. It should always be specified, which reflectance is meant

p11, I.5: traditional RTMs required over 10 minutes to run -> this depends very much on the settings, on the radiative transfer solution method etc. Specify, which kind of RTMs you refer to.

References:

Kox, S.; Bugliaro, L.; Ostler, A., Retrieval of cirrus cloud optical thickness and top altitude from geostationary remote sensing ATMOSPHERIC MEASUREMENT TECH-NIQUES Volume: 7 Issue: 10 Pages: 3233-3246 Published: 2014

СЗ

Loyola, Diego G. R.; Pedergnana, Mattia; Garcia, Sebastian Gimeno Smart sampling and incremental function learning for very large high dimensional data NEURAL NET-WORKS Volume: 78 Special Issue: SI Pages: 75-87 Published: JUN 2016

Efremenko, Dmitry S.; Loyola, Diego G.; Hedelt, Pascal; et al. Volcanic SO2 plume height retrieval from UV sensors using a full-physics inverse learning machine algorithm INTERNATIONAL JOURNAL OF REMOTE SENSING Volume: 38 Supplement: 1 Pages: 1-27 Published: 2017

Strandgren, Johan; Fricker, Jennifer; Bugliaro, Luca, Characterisation of the artificial neural network CiPS for cirrus cloud remote sensing with MSG/SEVIRI ATMO-SPHERIC MEASUREMENT TECHNIQUES Volume: 10 Issue: 11 Pages: 4317-4339 Published: NOV 14 2017

Loyola, Diego G.; Garcia, Sebastian Gimeno; Lutz, Ronny; et al. The operational cloud retrieval algorithms from TROPOMI on board Sentinel-5 Precursor ATMOSPHERIC MEASUREMENT TECHNIQUES Volume: 11 Issue: 1 Pages: 409-427 Published: JAN 18 2018

---

## Author Comment (AC1) · 15 Mar 2019

Many thanks to both reviewers for the helpful comments and for their time reviewing this work. They each identified several fundamental issues that were not clearly addressed in the original manuscript, and their input has substantially improved the quality of this work.

The primary changes we made to address the concerns of both reviewers are summarized below.

- We updated section 1 to more clearly state our novel contributions and their relation to prior work. - We added additional content to section 2 that formally describes how we train and validate the channelwise subnetworks to construct the neural RTM. -

[Figure]

We added additional content to section 3 that states our modeling assumptions and the specific parameters used to train the channelwise subnetworks. - We provide a commented LibRadTran config file associated with the PRISM flightline we considered in our case study as a supplement, citing the references therein in detail in the narrative. - We reorganized the paper in a manner that makes interpreting the content in the provided figures and tables easier to understand.

We provide an updated manuscript (nnrt_amt_r1.pdf) and detailed responses to individual referee comments (nnrt_amt_r1_responses.pdf) and also provide the libradtran configuration file used in our model runs (to be added as a supplemental file) in the attached zip file.

Our responses to referee comments are highlighted in red text, as are our modifications to the text and figures in the revised manuscript.

Please also note the supplement to this comment:
https://www.atmos-meas-tech-discuss.net/amt-2018-436/amt-2018-436-AC1-supplement.zip
* * *

---

## Author Comment (AC2) · 15 Mar 2019

Many thanks to both reviewers for the helpful comments and for their time reviewing this work. They each identified several fundamental issues that were not clearly addressed in the original manuscript, and their input has substantially improved the quality of this work. The primary changes we made to address the concerns of both reviewers are summarized below. - We updated section 1 to more clearly state our novel contributions and their relation to prior work. - We added additional content to section 2 that formally describes how we train and validate the channelwise subnetworks to construct the neural RTM. - We added additional content to section 3 that states our modeling assumptions and the specific parameters used to train the channelwise subnetworks. - We provide a commented LibRadTran config file associated with the PRISM flightline

we considered in our case study as a supplement, citing the references therein in detail in the narrative. - We reorganized the paper in a manner that makes interpreting the content in the provided figures and tables easier to understand.

We provide an updated manuscript (nnrt_amt_r1.pdf) and detailed responses to individual referee comments (nnrt_amt_r1_responses.pdf) and also provide the libradtran configuration file used in our model runs (to be added as a supplemental file) in the attached zip file.

Our responses to referee comments are highlighted in red text, and our modifications to the text and figures in the revised manuscript are shown in blue text.

Please also note the supplement to this comment:
https://www.atmos-meas-tech-discuss.net/amt-2018-436/amt-2018-436-AC2-supplement.zip

———————————————

---

## Author Response (AR2)

Response: We'd like to thank the associate editor for identifying a few corrections we missed in our revised manuscript. We have made the appropriate changes to the final version of the manuscript to correct these issues.

Also, thanks to both reviewers for the helpful comments and for their time reviewing this work. They each identified several fundamental issues that were not clearly addressed in the original manuscript, and their input has substantially improved the quality of this work.

The primary changes we made to address the concerns of both reviewers are summarized below.

- We updated section 1 to more clearly state our novel contributions and their relation to prior work.
- We added additional content to section 2 that formally describes how we train and validate the channelwise subnetworks to construct the neural RTM.
- We added additional content to section 3 that states our modeling assumptions and the specific parameters used to train the channelwise subnetworks.
- We provide a commented LibRadTran config file associated with the PRISM flightline we considered in our case study as a supplement, citing the references therein in detail in the narrative.
- We reorganized the paper in a manner that makes interpreting the content in the provided figures and tables easier to understand.

Our responses to the reviewer comments below are highlighted in red text. Our modifications to the text and figures are shown in blue text in the copy of the manuscript in this document that follows our responses.

**Anonymous Referee #1**

This paper is a compact presentation of a novel method of the emulation and use of radiative transfer models in compensating for the effects of illumination and the atmosphere to retrieve surface reflectance from measured radiances collected by an imaging spectrometer. It is quite dense and while it presents an adequate description, the results are just limited examples of the accuracy and a demonstration of its use.

It would be nice to see an expanded treatment of the first part of the article, the development of the forward emulation model, which is the main contribution. Section 2 presents the model with many details presented in a dense manner. It would be nice to expand and include more explicit equations for the various steps involved. More discussion on the selection and justification of the approach would also provide more insight.

Response: We added a new section that provides a comprehensive walkthrough of the entire neural RTM training procedure, and also describes the sampling/validation strategy we used to guide the optimization of the channelwise subnetworks.

We also provided more detailed description of the modeling assumptions made and parameters used to train each subnetwork used in our PRISM case study in sections 2 and 3.

In Section 3, the rationale behind limited the parameters of Table 1 would be good to see.

Response: We selected the suite of state parameters to act as a representative set of the types of states observed in operational atmospheric correction settings involving imaging spectroscopy data.

Our updates to the last paragraphs on page 7 clarify and motivate our parameter selection rationale.

In Figure 1, is the rho in the middle of the figure supposed to rho_obs? If so please label as such. If not, how does it relate to terms defined in the text?

Response: We added a revised version of Figure 1 that corrects the notational errors in the earlier version, and provided a more descriptive caption for the content shown in the figure.

Figure 5 should have a color legend to better interpret the quantitative results. Also, why show mean squared error? Why not root mean square which can be better interpreted?

Response: The error shown in Figure 5 was actually RMSE, but was incorrectly labeled in the earlier draft. We modified the caption and corresponding text to remedy this issue.

The labels that overlap the contours give the RMSE error level associated with each contour, but we also added a color bar to make this figure a bit easier to parse.

Actually, is this the mean error across the wavelengths? That would mask errors that may be concentrated in the water vapor region. Please clarify.

Response: The RMSE error depicted in Figure 5 is indeed averaged across all wavelengths. However, as we mentioned in our analysis, the high error values associated with increasing water vapor concentrations shown in the $H_2O$ row/column of the figure suggest that errors concentrated in the water absorption bands are not suppressed.

What was the source for the smoothed reflectance shown in Figure 6? Field spectrometer measurements? A library? Did you also use a dark target for the vicarious procedure?

Response: The radiance and the smoothed reflectance spectra shown in Figure 6 correspond to the same PRISM beach sand target. We adjusted the figure caption to clarify this detail.

This work presents an exciting development in the operational use of imaging spectrometers and deserves a more comprehensive presentation. Perhaps this could be the subject of future articles.

Response: thank you!

**Anonymous Referee #2**

The paper presents a neural network developed to calculate radiative transfer for the solar spectral region in clear sky. The model is applied to retrieve the surface spectral reflectance function from PRISM (airborne imaging spectrometer) data. As examples the retrieved surface reflectance spectra for 5 different surface types are shown to demonstrate that the method works well. The neural network method highly accelerates the atmospheric correction methodology, thus it is certainly an interesting approach which might become standard since the data to be processed increases rapidly with improvements in spatial and spectral resolution of sensors. The topic of the paper fits well in the scope of AMT, however, it needs to be revised, because the methodology needs to be described more precisely. Further the authors need to point, what makes their approach novel compared to other neural networks based approaches. I recommend publication in AMT after these revisions.

Major comments:
1. The application of neural networks for remote sensing has become rather common in the last decade (besides the mentioned references e.g. Kox et al. 2014, Loyola et al. 2016, Eferemeko et al 2017, Strandgren et al. 2017, Loyola et. al 2018). The authors say that they for the first time apply a neural network based algorithm to imaging spectroscopy. I am not sure whether this is correct. At least, the operational cloud retrieval algorithm for TROPOMI, a spaceborne imaging spectrometer, also applies a neural network RTM (Loyola et al., 2018). So in my opinion the authors should point out, what makes their approach novel and what is the difference to other similar approaches.

Response: We do not claim that this work represents the first time a neural network has been applied to imaging spectroscopy. Rather, we believe that this work represents the first attempt to learn the RTM function F(x) → y such that the trained RTM emulator is capable of acting as a forward model in an atmospheric correction procedure, allowing us to retrieve surface reflectance over the entire VSWIR range in variable imaging conditions.

Also, thank you for pointing us to Loyola 2018, which represents the most similar approach to ours. They perform RTM emulation as part of an inversion (as we do), but their approach is designed for atmospheric measurements (i.e., cloud height and fraction) rather than surface retrievals spanning multiple wavelengths.

We rewrote the last two paragraphs in the introduction to clarify these details.

If I understood correctly, the physical knowledge that is incorporated is the "analytical decomposition of the radiance spectrum into quantities that are individually easier to model". Later I find, that this means to use reflectances ("pi*y/phi0*E0" with y-radiance, phi0 solar zenith angle and E0 -extraterrestrial irradiance; apart from E0 very uncommon nomenclature) instead of radiance. This is a very common approach, also in look-up-table based retrieval algorithms and not a new idea. I think that all publications mentioned below use reflectances and not radiances.

Response: Thank you for identifying this discrepancy. The quoted sentence was indeed poorly written and contained a rather egregious typo, and we did not intend to insinuate that using reflectances instead of radiances was a novel component of this work. Specifically:

"…analytical decomposition of the radiance spectrum into quantities that are individually easier to model…"

should be

"analytical decomposition of the radiative transfer function F(x) into quantities that are individually easier to model…"

The above description refers to our approach to model the individual wavelengths of spectra generated by a first-principles RTM using separate neural network models. Using TOA reflectance rather than radiance to train a neural network is not (in itself) novel, the primary contribution is the channelwise decomposition of F(x) that is used for predicting TOA responses at multiple wavelengths.

We believe that the updates we made to the last two paragraphs in the introduction address this ambiguity.

The second "novel" idea that is mentioned is the channel-wise training. This is as far as I know already applied in Kox et al. 2014. The only new idea, that I have not seen so far, is the channelwise weight propagation, which makes sense to me.

While it is indeed true that Kox et al. (and many others) train neural networks using multiple channels as input, those approaches are quite different from ours, both in terms of network architecture and derived outputs. Specifically:
-   We train $k$ separate neural networks, where each subnetwork computes the mapping from the state space to TOA reflectance for a single wavelength. The combined output of all k networks yields a full TOA reflectance spectrum, and allows us to directly apply the trained neural RTM to act as a forward model in an RTM-based atmospheric correction procedure.

- In contrast, Kox et al. construct a single neural network using a combination of *k* radiance channels + several auxiliary parameters to derive two atmospheric parameters (cloud optical thickness and top altitude).

To more clearly distinguish our approach from prior work, we added Algorithm 1 and several new paragraphs describing the end-to-end training + validation methodology we used.

2. The setup of the neural network needs a more comprehensive description:
- Most symbols in Figure 1 are not explained

Response: We added a revised version of Figure 1 that corrects the notational errors in the earlier version, and provided a more descriptive caption for the content shown in the figure.

- It is not clear to me, whether the "monochromatic subnetworks" (p4, l.4) are really monochromatic (for 1 wavelength) or for one channel (convolved with the spectral response function). E.g. on p5, l4 it reads "To ensure each subnetwork reliably models its corresponding channel ..."

Response: In principle, the neural RTM is designed to compute the mapping from the state space for each wavelength of the TOA reflectance spectra generated by the RTM (LibRadTran). However, the neural RTM we demonstrate in this work generates TOA reflectance spectra at (PRISM) instrument wavelengths, and the LibRadTran TOA spectra are indeed resampled with respect to the PRISM spectral response function to match those wavelengths. Consequently, they are not strictly monochromatic, but we decided that the dramatic reduction in computation made this an acceptable compromise for this proof of concept, and plan to construct a more general neural RTM at the same spectral resolution as the LibRadTran outputs in future work.

Our updates on p9, 1st paragraph describe these modeling assumptions + the associated benefits compromises of this approach in detail.

- Explain settings: "feed-forward architecture with two hidden layers of 50 units each ... Why is the neural network set up as shown in Figure 1? Why are these particular settings chosen for this particular application? Explain terms "rectifying" and "early stopping"

Response: Rectifying linear unit (RELU) activations were introduced by [Nair, 2010] and simply involve clipping the negative portion of the input (i.e., $f(x) = \max(0,x)$). RELU activations have been shown to be more robust than the conventional sigmoid / tanh activations used historically in neural networks. We updated the text to state this detail.

Early stopping the process of training each neural network until some criteria are met, for instance when test accuracy plateaus for several cycles. We removed any references to this item, as we explain our stopping conditions in more detail in section 3.

Our updates starting on p9, 2st paragraph describe the specific parameters we used to train each subnetwork, along with context motivating our design decisions.

3. The radiative transfer simulations for training and validation were performed using the libRadtran package, which is a comprehensive package including various parameterizations, atmospheric setups, aerosol types etc. The setup of libRadtran needs to be specified, i.e. which atmosphere is used, which absorption parameterization, how is the aerosol defined, which radiative transfer solver is used etc.

Response: We provided a commented libradtran configuration file (prm20151026t173148_libradtran_config) associated with the PRISM flightline described in our case study as a supplemental file, and describe the atmosphere used in the last paragraph of page 7.

The symbols in Table 1 (input parameters) are also not defined.

Response: the symbols in Table 1 were defined in the first paragraph of Section 2, but we agree that the table did not clearly reference the associated symbols. We moved this paragraph closer to the table, and added text labels for each symbol listed in the table for easier interpretation.

4. Figure 3: How is the linear regression done? Linear interpolation in a lookup-table for the parameters listed in Table 1? This is not explained in the text.

Response: We trained a single least squares linear regressor for each channel using the same set of training samples used to train the subnetwork associated with that channel. We then computed the test error per channel for each of trained regressor on the same set of test samples as the subnetwork. The channelwise test errors using the linear regressors provide an approximate upper bound of the error that would be incurred using piecewise / locally linear interpolation to infer TOA responses for intermediate states based on lookup tables.

These details are reflected in our updates to the first paragraph on page 10.

5. Why are only 5 examples of retrieved surface reflectivity spectra shown. Since the method is so fast, it could be easily be applied on a full image. The difference between the traditional atmospheric correction approach and the new neural network approach should be assessed in a statistical sense.

Response: Our focus in this work was to provide a detailed procedure to construct an RTM emulator that can be used as a forward model yielding high quality surface reflectance retrievals. To that end, we felt a qualitative analysis of our reflectance retrievals on spectra for which the surface materials were known was the most important component to demonstrate a

functional proof of concept. We are currently applying the methodology described in this work in operational settings for entire images, and plan to describe those results in more detail in future publications.

Minor comments:
- Figure 5: I suggest to create a separate figure for the histograms shown in the diagonal with proper y-axis labels.

Response: Removing the histograms from the diagonal would leave large empty areas with no useful content, and we believe that associating each parameter on the diagonal with the error surfaces for the other parameters in each column provides informative context.

However, your point regarding the disconnect between the range of error values for the histograms and the error surfaces is well taken. To address this, we added text labels to the diagonal subplots specifying the minimum + maximum error values and placed those labels on the bar of the corresponding histogram bin in Figure 5.

- It is confusing that the term reflectance is used for the top-of-atmosphere reflectance and the surface-reflectance. It should always be specified, which reflectance is meant

Response: We updated each reference to "reflectance" to specify whether we refer to "surface reflectance" or "TOA reflectance."

p11, l.5: traditional RTMs required over 10 minutes to run -> this depends very much on the settings, on the radiative transfer solution method etc. Specify, which kind of RTMs you refer to.

Response: Good point – The updates we adde on page 9 specify the assumptions we made with respect to the RTM model, and we provided additional information in the last paragraph of page 11 regarding the runtime of our model versus the first-principles RTM. In particular, we noted the runtime of each approach with respect to the number of wavelengths in the output spectrum. Also, the libradtran config file we provide as a supplement should provide context regarding the RTM we used in this work.

[revised manuscript text omitted]